# Contemporary Drug Therapy for Renal Cell Carcinoma—Evidence Accumulation and Histological Implications in Treatment Strategy

**DOI:** 10.3390/biomedicines10112840

**Published:** 2022-11-07

**Authors:** Kazutoshi Yamana, Riuko Ohashi, Yoshihiko Tomita

**Affiliations:** 1Department of Urology and Molecular Oncology, Niigata University Graduate School of Medical and Dental Sciences, 1-757 Asahimachi-Dori, Chuo-Ku, Niigata 951-8510, Japan; 2Division of Molecular and Diagnostic Pathology, Niigata University Graduate School of Medical and Dental Sciences, 1-757 Asahimachi-Dori, Chuo-Ku, Niigata 951-8510, Japan

**Keywords:** renal cell carcinoma, drug therapy, histological subtypes, immunotherapy, biology-driven treatment strategy

## Abstract

Renal cell carcinoma (RCC) is a heterogeneous disease comprising a variety of histological subtypes. Approximately 70–80% of RCC cases are clear cell carcinoma (ccRCC), while the remaining subtypes constitute non-clear cell carcinoma (nccRCC). The medical treatment of RCC has greatly changed in recent years through advances in molecularly targeted therapies and immunotherapies. Most of the novel systemic therapies currently available have been approved based on ccRCC clinical trial data. nccRCC can be subdivided into more than 40 histological subtypes that have distinct clinical, histomorphological, immunohistochemical, and molecular features. These entities are listed as emerging in the 2022 World Health Organization classification. The diagnosis of nccRCC and treatments based on cancer histology and biology remain challenging due to the disease’s rarity. We reviewed clinical trials focused on recent discoveries regarding clinicopathological features.

## 1. Introduction

Approximately 70–80% of clear cell carcinoma (RCC) cases are diagnosed as clear cell carcinoma (ccRCC); the remaining subtypes are categorized as non-clear cell carcinoma (nccRCC). Accordingly, the development of new drugs has focused on ccRCC, with little attention given to nccRCC. The prognosis of patients with metastatic renal cell carcinoma (mRCC) has improved with the use of immuno-oncology (IO) agents compared to that achieved through the use of cytokines [1] and molecularly targeted therapy (TT) [2,3,4]. Novel systemic therapies are now available, having been approved based on ccRCC clinical trial data. Patients with nccRCC have few evidence-based treatment options and tend to have poor prognoses. As a disease type, nccRCC includes various histological subtypes with distinct clinical and biological characteristics. Herein, we review the pivotal clinical trials for advanced RCC therapy, organized by histological types, and highlight avenues for further research.

## 2. Evidence from Clinical Trials

Generally, systemic therapy for RCC can be categorized into three eras: cytokine therapy, target therapy (mainly targeting vascular endothelial growth factor (VEGF) or Mammalian target of rapamycin (mTOR)), and IO therapy. Novel biological data could lead new target options in some subtypes. We defined these new-generation TTs as biology-driven therapies in this review (Figure 1).

### 2.1. Pre-Immuno-Oncology Drugs

#### 2.1.1. Clear Cell Renal Cell Carcinoma (ccRCC)

##### Cytokine Era

Interferon-alpha (IFNα) and interleukin-2 (IL-2), which represent first-generation immunotherapy, are no longer first-line systemic therapies for advanced RCC. However, before the target therapy era, these agents were the only treatments available [5,6,7] (Table 1). Cytokine therapy could be beneficial for selected patients with a favorable risk and metastatic organs, such as those only with lung metastasis [8]. We have previously reviewed data from 22 patients treated with cytokine therapy (IFNα and/or low-dose IL-2) as a first line therapy in the target therapy era (between 2006 and 2018). The median overall survivals (OS) were 123 months, 36 months, and 16 months in favorable, intermediate, and poor International Metastatic Database Consortium (IMDC) subgroups [2], respectively (*p* < 0.01). Anemia and high C-reactive protein levels are risk factors for poor prognosis (*p* = 0.022 and *p* = 0.042, respectively). The IMDC-favorable risk group with only lung metastasis and a C-reactive protein level within the normal range is considered a suitable candidate for cytokine therapy (data unpublished). Some patients treated with cytokine therapy may achieve a good long-term response at a relatively low cost, even though this approach is no longer considered a first-line treatment. Moreover, the self-infection of anti-cancer agents may help preserve a good quality of life.

##### Target Therapy Era

Since December 2005, several molecular therapy agents have been approved for the treatment of advanced RCC [9,10,11,12,13,14,15,16,24]. ccRCC is associated with a mutation or inactivation of the *von Hippel- Lindau* (*VHL*) gene and the resultant over-expression of vascular endothelial growth factor (VEGF) [25]. The first drug to target VEGF in the treatment of ccRCC was the monoclonal antibody bevacizumab [24]. In addition, multi-targeted tyrosine kinase inhibitors (TKI) including sorafenib, sunitinib, pazopanib, axitinib, cabozantinib, and lenvatinib are currently in use [9,10,11,12,13,14]. Mammalian target of rapamycin (mTOR) is the second validated therapeutic target used by the inhibitors temsirolimus and everolimus [15,16]. For more than a decade, sequential treatment with target agent monotherapy has been the leading approach to treatment, helping to improve survival. However, for first-line therapies, the response duration is estimated at 12 months; subsequently, treatment resistance may develop, highlighting a need for alternative strategies [13,14].

#### 2.1.2. ccRCC with Sarcomatoid and/or Rhabdoid Differentiation

All RCC types involving sarcomatoid or rhabdoid features are associated with poor prognosis [26]. In ccRCC, these features reduce the efficacy of target therapies such as VEGF-TKI and mTOR-I. A phase II single-arm trial of sunitinib and gemcitabine in patients with sarcomatoid features aimed to assess the impact of including cytotoxic agents in the regimen. The Overall Response Rate (ORR) was 26%, with median Time to Progression (TTP) and OS rates of 5 months and 10 months, respectively. The combination might be more efficacious than either therapy alone; however, it is not as valuable in RCC patients without sarcomatoid features [27,28] (Table 2).

#### 2.1.3. Non-ccRCC

##### Papillary RCC (pRCC)

Cytokine era

Papillary RCC (pRCC) is the most common subtype of nccRCC, and some evidence regarding cytokine efficacy in this context is available. The Program Etude Rein Cytokines (PERCY) Quattro trial investigated IFNα, IL-2, medroxyprogesterone, and a combination thereof in the treatment of this disease. Patients with various RCC histology types, including nccRCC, were randomized in a two-by-two factorial design. No objective response was observed in the nccRCC patients with papillary (*n* = 21), chromophobe (*n* = 4), collecting duct (*n* = 1), or sarcomatoid (*n* = 3) subtypes. No evidence of a survival benefit was reported [34] (Table 3). 

Target therapy era: target VEGF or mTOR

SWOG 1107 was an RCT dedicated to pRCC patients (*n* = 50). The following study compared VEGF-TKI tivantinib with or without EGFR-TKI erlotinib as a first- and second-line treatment. The median PFS (2.0 months vs 5.4 months) and OS (10.3 months vs 11.3 months) were comparable in both arms [35]. In the Phase III Advanced Renal Cell Carcinoma (ARCC) trial, high-risk RCC patients with various histological types were randomized to receive IFNα or mTOR inhibitor temsirolimus. A subgroup analysis of the outcomes for nccRCC (*n* = 37), which included mostly patients with pRCC, showed a median PFS (mPFS) of 7.0 months and a median OS (mOS) of 11.6 months for temsirolimus; the corresponding values for IFNα were 1.8 months and 4.3 months, respectively. The disease control rate (DCR) was reported in 41% and 8% of the patients receiving temsirolimus and IFNα, respectively. Temsirolimus was more effective at improving the PFS and OS rates than IFNα in patients with nccRCC [36]. Two principal RCTs have compared everolimus to sunitinib as a first-line treatment for advanced nccRCC with various histological types. In the ASPEN trials [37], pRCC (*n* = 70 of 109) was associated with an ORR of 5% and an mPFS of 5.5 months for everolimus, and an ORR of 24% and mPFS of 8.1 months for sunitinib. The OS estimates were not reported. In the ESPN trial [38], pRCC (*n* = 27 of 72) was associated with an mPFS of 4.1 and mOS of 14.9 months for everolimus; the corresponding values for sunitinib were 5.7 months and 16.6 months, respectively. Overall, sunitinib was preferred over everolimus as a first-line treatment in pRCC. Furthermore, several single-arm phase II trials involved patients with papillary histology alone. The RAPTOR trial (*n* = 50), which evaluated everolimus as a first-line treatment, resulted in an mPFS of 4.1 months and an mOS of 21.4 months. The SUPAP trial (*n* = 61) evaluated sunitinib as a first-line therapy for pRCC in two patients with either type 1 or type 2 disease; the mPFS rates were 6.6 months and 5.5 months, and the mOS rates were 17.8 months and 12.4 months, respectively. Both trials showed slight differences, and both sunitinib and everolimus remain used as a first-line treatment [39,40]. A separate trial evaluated a combination therapy with everolimus plus bevacizumab as a first-line treatment, which resulted in an ORR of 29% for nccRCC. In a subgroup analysis (*n* = 18 of 34), compared to disease types other than pRCC, pRCC including papillary features of unclassified RCC (uRCC; named RCC NOS by the WHO in 2022) was associated with an ORR of 43% vs 11%, and an mPFS and mOS of 12.9 vs 1.9 and 28.2 vs 9.3 months, respectively [41,42]. Recently, a phase II study of first-line lenvatinib plus everolimus in nccRCC reported that in pRCC (*n* = 20 of 31), the ORR and DCR were 15% and 85%, with mPFS and OS rates of 9.2 months and 11.7 months, respectively [43].

Biology-driven era; target MET

*MET* is a well-documented alteration in pRCC. MET inhibitors are potential treatments for diseases with papillary histology. Foretinib is a dual MET/VEGFR2-targeting inhibitor. In a phase II study (*n* = 67), patients treated with first- and second-line foretinib had an mPFS of 9.3 months (mOS not reached). The associated ORR estimates were in the range from 9% to 50% without and with a germline mutation in *MET*, respectively [47]. A phase II trial (*n* = 109) involving savolitinib, which is a highly selective MET inhibitor, as an any-line treatment for pRCC, reported an ORR of 18% for MET-driven disease but an absent value for MET-independent disease. Meanwhile, the mPFS rates for patients with MET-driven and MET-independent pRCC were 6.2 months and 1.4 months, respectively [48]. Crizotinib is a TKI that targets MET in addition to ALK and ROS1. The CREATE (*n* = 23) trial with any-line treatments with crizotinib for pRCC type 1 showed a two out of four (50%) PR in MET alteration patients but a 6.3% ORR in MET wild-type patients [44]. These results suggest that the molecular characterization of *MET* status was a better predictive marker of a response to MET inhibitors in pRCC. In 2020, the results of SAVOIR phase III RCT were published [45]. This trial involved pRCC patients with *MET*-driven tumors (chromosome 7 gain, *MET* or *HGF* amplification, or *MET* kinase mutation) randomized in a one to one ratio to receive either savolitinib or sunitinib. Only 60 patients met the criteria because of a lack of *MET*-driven alterations even though 254 patients were screened. The mPFS rates were 7.0 months and 5.6 months with savolitinib and sunitinib (Hazard Ratio (HR), 0.71; *p* = 0.313), respectively. The mOS was not reached for the patients on savolitinib while it was 13.2 months for sunitinib (HR, 0.51; *p* = 0.110). The ORR estimates were 27% and 7% for savolitinib and sunitinib, respectively, with some evidence of better efficacy and lower toxicity in savolitinib than in sunitinib [48]. In 2021, the SWOG1500 PAPMET phase II trial reported results that may change the standard of care for advanced pRCC. Cabozantinib was evaluated alongside sunitinib, savolitinib, and crizotinib in primarily treatment-naïve patients with pRCC (*n*  =  147). The PFS estimates were better for cabozantinib than for sunitinib (median 9.0 vs 5.6 months; HR, 0.60; *p* = 0.019), with the corresponding ORR estimates of 23% and 4%, respectively. Savolitinib and crizotinib were removed from the trial due to poor outcomes. The mOS was 20.0 months for cabozantinib and 16.4 months for sunitinib. These results are consistent with those of the CABOSUN trial, which applied the same randomization in ccRCC [46].

##### Chromophobe RCC (chRCC)

Cytokine era

Metastatic chromophobe RCC (chRCC) is a very rare disease; consequently, it has received no dedicated trials. The PERCY Quattro trial investigated patients with various RCC histology types including nccRCC. No objective response was observed in the evaluable patients with chRCC (*n* = 4) [34].

Target therapy era

In general, chRCC is an indolent subtype of RCC; however, 5–10% of patients with progressing disease have poor outcomes [61]. No RCT has been dedicated to chRCC. However, the ASPEN and ESPN trials included chRCC patients (Table 3). The ASPEN trial (*n* = 16 of 109) reported an mPFS of 11.4 months for everolimus and that of 5.5 months for sunitinib (OS data not shown). The ESPN (*n* = 12 of 72) trial reported the mPFS as not reached and an mOS of 25.1 months for everolimus; the corresponding values for sunitinib were 8.9 months and 31.6 months, respectively. These results suggest slight PFS benefits for everolimus, although the differences were not significant. Recent findings from a phase II trial of first-line lenvatinib plus everolimus in nccRCC are available. In patients with chRCC (*n* = 9 of 31), the ORR and DCR were estimated at 44% and 78%, respectively, with an mPFS of 13.1 months (mOS not reached) [43]. 

Biology-driven era

The genetic basis of sporadic chRCC remains limited; consequently, no RCT has been conducted to date.

##### Collecting Duct Carcinoma (CDC)

Chemotherapy

Collecting duct carcinoma (CDC) is an aggressive subtype; however, there have been no dedicated RCTs to date. CDC behaves like a more aggressive urothelial cancer type rather than RCC. Accordingly, a commonly used medical treatment for advanced CDC is platinum-based chemotherapy. There have been three single-arm phase II trials concerning traditional chemotherapy in this context. First, a study with gemcitabine plus cisplatin or carboplatin (*n* = 23) reported an ORR of 26%, with an mPFS of 7.1 months and an mOS of 10.5 months [54]. Second, a study with VEGF-TKI sorafenib in combination with gemcitabine and cisplatin (*n* = 26) reported an ORR of 30.8%, a DCR of 84.6%, an mPFS of 8.8 months, and an mOS of 12.5 months [55]. Third, the BEVAEL trial using bevacizumab plus gemcitabine and platinum in CDC and SMARCB1-deficient renal medullary carcinoma (RMC) reported an ORR of 39% with an mOS of 11 months (*n* = 26 of 34). Based on these results, gemcitabine and cisplatin regimens, without the addition of other agents, remain the standard treatment for patients with CDC; nevertheless, the outcomes remain poor [56].

Target therapy era

A phase II trial of sunitinib for nccRCC (*n* = 6 of 57) included the CDC subtype, reporting an ORR of 0% and an mPFS of 3.1 months [57]. In addition, the BONSAI (*n* = 23) phase II single-arm trial of cabozantinib as a first-line treatment for mCDC was presented at ESMO 2021, reporting an ORR of 35% with an mPFS of 4 months and mOS of 7 months. The authors concluded that cabozantinib has promising efficacy and acceptable tolerability in mCDC patients [58].

##### TFE3- and TFEB-Rearranged RCCs (Formerly Microphthalmia Transcription Factor (MiT) Family Translocation RCC (tRCC) in WHO2016)

tRCCs are very rare tumors that are more aggressive in adults [61,62]. No RCTs have focused on these patients. Some retrospective studies have shown a modest response to target therapy [63].

##### Fumarate Hydratase (FH)-Deficient RCC

Hereditary leiomyomatosis and renal cell cancer (HLRCC) is familial cancer syndrome associated with an aggressive RCC type, which is caused by germline *FH* mutation. Sporadic *FH* mutations can also occur [61,64]. An *FH* mutation may inactivate the enzyme and change the function of tricarboxylic acid. A phase II study of bevacizumab and erlotinib enrolled a total of 83 patients with pRCC undergoing first- and second-line treatments (AVATAR trial); the sample was split approximately evenly between HLRCC and sporadic papillary RCC. HLRCC was associated with an ORR of 64% and a PFS of 21.1 months; the corresponding values for pRCC were 35% and 8.8 months, respectively. This regimen may be a suitable option for a select population [53].

##### SMARCB1-Deficient Renal Medullary Carcinoma (RMC)

Chemotherapy

RMC is a rare RCC type characterized by the loss of tumor suppressor SMARCB1 and high mortality rates. No RCT has focused on this subtype; however, several retrospective studies have been reported. RMC does not respond to TKIs; thus, platinum-based chemotherapy such as carboplatin plus paclitaxel is the preferred first-line therapy. Nevertheless, the associated response rate remains at 29% and the response duration tends to be brief (*n* = 52). As a second-line treatment, gemcitabine plus doxorubicin (*n* = 16) has shown some clinical activity in patients with platinum-refractory RMC (ORR, 18.8%; PFS and OS of 2.8 months and 8.1 months, respectively) [59,60]. 

Biology-driven era—target EZH2

An EZH2 inhibitor termed tazemetostat was recently approved by the Food and Drug Administration (FDA) for the treatment of another SMARCB1-deficient malignancy, namely, epithelioid sarcoma [65]. A phase II trial (NCT02601950) involved 14 patients with RMC and one patient with RCCU-MP; however, enrolment in the tazemetostat trial has been suspended due to safety concerns. The loss of SMARCB1 may induce proteotoxic and replication stress; thus, a proteasome inhibitor is a potential therapeutic agent. A phase II clinical trial (NCT03587662) is evaluating the combination of the proteasome inhibitor ixazomib with gemcitabine and doxorubicin in RMC [66]. The results of these studies are forthcoming.

## 2.2. Immuno-Oncology drugs

### 2.2.1. ccRCC

The CheckMate 025 and CheckMate 214 studies, pivotal phase III studies of nivolumab (NIVO) and nivolumab + ipilimumab (NIVO + IPI), reported a significant OS benefit with a moderate improvement in PFS. In addition, several IO + VEGF or TKI combination therapy regimens were approved after the publication of results from pivotal phase III trials such as KEYNOTE 426, JAVELIN Renal 101, Immotion 151, Checkmate 9ER, and CLEAR. Most of these regimens replace sunitinib, which had previously been the standard treatment (Table 1). Currently, we are experiencing the era of IO combination therapy (IO combo). The latest ESMO guidelines [67] for all risk groups recommend an IO combo as a first-line therapy. Meanwhile, the OS signals in favorable risk patients are still immature and not yet superior to sunitinib.

#### IO Monotherapy

CheckMate 025 has shown superior efficacy for nivolumab over everolimus in patients with ccRCC previously treated with one or two antiangiogenic regimens with improved safety and tolerability [68]. Concurrently, the use of an IO re-challenge following other IO combination therapies is considered an experimental approach rather than the standard of care.

#### IO Doublet

The CheckMate 214 data demonstrated a survival benefit for patients treated with nivolumab and ipilimumab compared with those treated with sunitinib in intermediate/poor risk advanced RCC, ushering in the front-line IO era. Data from a 5-year follow-up study on an IO doublet showed a durable clinical benefit with NIVO + IPI, suggesting that patients that show a disease response and remain alive at the 3-year mark may continue with those outcomes at the 5-year mark, presenting a plateau in the tails [17,18]. An IO-doublet rechallenge following IO monotherapy is an experimental approach associated with a lower response rate than expected at this point [69].

#### IO + VEGF Inhibition

In the KEYNOTE-426 study, pembrolizumab + axitinib showed significant improvement in terms of OS, PFS, and ORR estimates, compared to sunitinib as a first-line therapy for advanced RCC. The JAVELIN Renal 101 (avelumab + axitinib), Immotion 151 (atezolizumab + bevacizumab), Checkmate 9ER (nivolumab + cabozantinib), and CLEAR (pembrolizumab + envatinib) trials also showed a better efficacy compared to sunitinib [19,20,21,22,23]. However, no head-to-head study has compared these regimens, and treatment decisions are made on a case-by-case fashion based on various factors.

#### Target Therapy in IO Era 

The role of target agents, specifically, monotherapy, is controversial. In first-line therapy, VEGF-TKI is recommended in combination with IO agents; it remains an acceptable alternative unless IO therapy is contraindicated or not available. Target therapy alone may be another option in selected patients, such as those presenting with low-volume, asymptomatic, and slow-growing disease. Conditionally, we should keep checking the longer follow-up mature data of OS signals with these favorable risk patients, as a recent report showed a loss of the long-term OS benefits observed in the KEYNOTE-426 study [70] 

As a subsequent therapy, target agents that have not been given are recommended. Robust prospective data after IO combo therapy are lacking, but a few prospective data and retrospective data support the expectations for sequencing therapy. In our data, with respect to the AFTER I-O study, patients that participated in CheckMate 025 or CheckMate 214 (*n* = 45) who received target agents after nivolumab with or without ipilimumab were analyzed retrospectively. The median PFS2 of NIVO and NIVO + IPI was 36.7 and 32.0 months, respectively. The median OS from first-line therapy was 70.5 months for patients treated with NIVO, while it was not reached with NIVO + IPI. The safety profile of each TT after each IO was similar to previous reports regarding the use of first-line therapies. These results indicate that sequential target therapy after IO may improve survival; nevertheless, these findings should be approached with caution, as they are from a small retrospective study [71,72]. A separate retrospective study assessed the clinical effectiveness of target therapy after IO therapy-treated patients who received VEGF-TKI had improved clinical outcomes with respect to mTOR inhibitor following IO therapy [73].

### 2.2.2. ccRCC with Sarcomatoid and/or Rhabdoid Differentiation

The CheckMate 214 post hoc analyses of nivolumab plus ipilimumab (NIVO + IPI) for ccRCC with sarcomatoid features showed promising results. During the minimum follow-up period of 42 months, the median OS was better for NIVO + IPI (not reached) than for sunitinib (14.2 months), with the corresponding HR of 0.45. The PFS estimates were also better for NIVO + IPI than for sunitinib (median 26.5 vs. 5.1 months; HR, 0.54). The reported ORR was 60.8% with NIVO + IPI vs 23.1% with sunitinib, with CR rates of 18.9% vs 3.1%, respectively. Furthermore, the other IO combination trials such as Keynote 426, Immotion 151, and JAVERIN Renai101 evaluated sarcomatoid RCC in subgroup analyses. The updated analysis of CheckMate 9ER and CLEAR trials evaluated outcomes stratified by sarcomatoid features in 2021, showing that an IO combo achieved better outcomes than sunitinib as a first-line treatment in advanced sarcomatoid ccRCC. Recent studies have suggested that the benefits of IO combo therapy may be associated with high genomic instability, an elevated T-effector signature, and higher PD-L1 expression and tumor mutational burden compared to those in RCC without sarcomatoid features. These results suggest that an IO combo may be a suitable first-line treatment for sarcomatoid differentiation in RCC [30,31,32,33] (Table 2).

### 2.2.3. Non-ccRCC

#### pRCC

The CALYPSO study was a phase II trial investigating the combination of MET and PD-L1 inhibition in advanced pRCC. This trial enrolled 41 patients who were either VEGF TKI-naïve or -refractory. The patients received both highly selected MET inhibitor savolitinib and anti-PD-L1 agent durvalumab. The ORR was 29%, with an mPFS and mOS of 4.9 months and 12.3 months, respectively. Among 14 patients with *MET*-driven tumors, the confirmed RR was 57% with a response duration of 9.4 months. Further, the mPFS and mOS with *MET*-driven tumors were 10.5 months and 27.4 months, respectively. The PFS was substantially longer for the patients with *MET*-driven than non-*MET*-driven tumors. This IO combo of savolitinib and durvalumab has encouraging clinical activity in patients with *MET*-driven pRCC [51]. In some RCTs for nccRCC with various histologies, pRCC was the most common subtype. Two RCTs with a single IO agent, anti PD-L1 pembrolizumab, or anti PD-L1 nivolumab showed inconsistent results. The phase II Keynote427 cohort B treated with first-line pembrolizumab included 165 nccRCC patients. Overall, 118 of the 165 pRCC patients showed ORR and DCR rates of 28.8% and 47.5%, respectively, compared to the corresponding values from the intention to treat (ITT) population of 26.7% and 43.0%, respectively. The phase III/IV Checkmate 374 trial with nivolumab enrolled 44 nccRCC patients, who received zero to three prior treatments, and reported an ORR of 8.3% and DCR of 50.0% in 24 of 44 pRCC patients compared to 13.6% and 50.0% reported in the ITT population. This discrepancy in the ORR rates between the two RCTs may be accounted for by the study’s eligibility criteria and the inclusion of pretreated patients [49,50]. A phase II trial (NCT03635892) of nivolumab and cabozantinib (IO combo) in patients with nccRCC is ongoing. MET and multi-targeted agent cabozantinib and IO drugs have shown favorable effects as monotherapies in pRCC. A combination therapy is expected to show synergistic effects. This trial enrolled 47 patients with advanced nccRCC who had received no prior systemic therapy or a single line of treatment of other-than-IO drugs. The patients were divided into Cohort 1 (*n* = 40; pRCC, tRCC, or uRCC) and Cohort 2 (*n* = 7; chRCC). The median follow-up period was 13.1 months, and the results were reported at ASCO 2021. In Cohort 1, most patients (*n* = 26, 65%) were previously untreated; meanwhile, 14 (35%) patients had one prior treatment with VEGF-TKI or mTOR-I. In this cohort, the ORR, DCR, PFS, and OS estimates were 47.5%, 97.5%, 12.5 months, and 28 months, respectively, suggesting this combination has promising efficacy and safety profiles in pRCC, tRCC, and uRCC [52] (Table 3). 

#### chRCC

Four trials involving IO drugs for nccRCC reported subgroup data for chRCC. The Keynote-427 cohort B using pembrolizumab (*n* = 21) as a first-line treatment showed an ORR of 9.5% and a DCR of 33.3%. The Checkmate 374 trial with nivolumab (*n* = 7) reported an ORR of 28.5% and a DCR of 85.7%. A third study using atezolizumab plus bevacizumab reported an ORR of 10%. A recent phase II trial (NCT03635892) of nivolumab and cabozantinib in patients with nccRCC cohort 2 included chRCC (*n* = 7), which showed no response. The ORR and DCR estimates were 0% and 71.4%, respectively (Table 3). 

In general, chRCC is a low-malignancy tumor type with a 5–10% risk of progression and metastasis. A multi-center re-evaluation study by Ohashi et al. suggested a new grading system based only on the presence of sarcomatoid differentiation and necrosis, which was an indicator of a limited response to treatment and poor prognosis [74]. In chRCC, mTOR-I and VEGF-TKI yielded responses comparable to those observed in other nccRCC subtypes, while I-O therapies combined with other agents did not improve outcomes regardless of having a better potential towards the sarcomatoid subtype [49,50,52,73].

#### tRCC

One retrospective study (*n* = 24) with various IO drugs used as a second-line treatment for metastatic tRCC reported ORR and DCR rates of 16.7% and 29.2%, respectively, with a PFS rate of 2.5 months. A recent retrospective analysis combining the IMDC and Harvard datasets reported ORR (25.0% with IO and 0% with TKI) and mOS (62.4 months with IO and 10.3 months with TKI) estimates. The authors concluded that IO therapy may be more beneficial than VEGF target therapy in tRCC [75,76].

#### RMC

Three trials are exploring the use of IO in patients with RMC. Most recently, a phase II trial (NCT03274258) enrolled RMC patients to assess the efficacy and safety of treatment with NIVO + IPI. Another phase II trial (NCT02721732) using pembrolizumab for rare tumors (*n* = 4 of 127) and a phase I study (NCT02496208) using cabozantinib and nivolumab alone or with ipilimumab for metastatic UC and other genitourinary tumors (*n* = 3 of 54) included some RMC patients. These data may help develop novel treatments, including IO therapies and biologic agents [77,78,79]. 

## 3. Updated Treatment Strategy—Noteworthy Clinical Trials

### 3.1. ccRCC

#### 3.1.1. 1st Line Therapy

##### Pegylated Interleukin-2 (IL-2)

Among cytokine therapies, high-dose IL-2 therapy provokes a durable response, but its treatment-related AE has limited its use [5]. Bempegaldesleukin (BEMPEG) is a PEGylated IL-2, a type of novel IL-2 receptor agonist, which is a stable fusion protein designed to activate and proliferate CD8+ T cells and NK cells. In a phase I study, BEMPEG was well-tolerated; combined with nivolumab, it showed a promising ORR (71%) and manageable toxicity in untreated RCC patients. A phase III PIVOT-09 trial is currently investigating BEMPEG + nivolumab vs sunitinib or cabozantinib (investigator’s choice) as a first-line treatment for advanced ccRCC. This trial aims to evaluate the ORR and OS in the IMDC intermediate/poor risk and ITT populations. The secondary aim of this study is to estimate the PFS in the IMDC intermediate/poor risk and ITT populations, as well as to evaluate its safety profile, associated PD-L1 expression (predictive biomarker), and patients’ quality of life [80,81]. In April 2022, it was announced that this study did not meet the prespecified threshold for statistical significance. The data have not been shared; however, a review and publication of the interim findings are expected.

##### Triple Combination

IO + IO or IO + TKI combination strategies have shown promising results; a trial to maximize their associated benefits is ongoing. COSMIC-313 is a phase III study evaluating the efficacy and safety of nivolumab + ipilimumab with or without cabozantinib in previously untreated patients with IMDC-intermediate or poor-risk aRCC. A prolonged PFS (HR 0.73) with triplet therapy was presented at ESMO 2022, and the secondary endpoint OS needs a further follow-up [82]. Another phase III study is comparing triple combinations (pembrolizumab + quavonlimab + Lenvatinib or pembrolizumab + belzutifan + Lenvatinib).

#### 3.1.2. Subsequent Therapy

##### IO–IO Combination

The phase II FRACTION-RCC/NCT02996110 trial involves a combination of NIVO + IPI after immunotherapy for patients with advanced RCC. The primary outcomes will be the ORR, DOR, and PFS rates. The secondary outcomes will include adverse events and serious adverse events. This trial will assess novel IO-IO combination therapy in patients with disease that was refractory to previous-line treatments [69].

##### IO–TKI Combination

IO + IO combo and IO + TKI combination may be feasible subsequent-line therapies post-frontline immunotherapy. The CONTACT-3 study is a phase III trial of cabozantinib with or without atezolizumab in several advanced RCC histology types, aiming to evaluate PFS and OS rates. The TiNivo-2 study of tivozanib with or without nivolumab aims to evaluate the PFS as the primary outcome, as well as the OS, ORR, and DOR rates and safety profile as the secondary outcomes [83,84].

##### HIF2α Inhibitor

Emerging agents are being designed to inhibit the transcription factor hypoxia-inducible factor (HIF), specifically, the HIF2α subunit. The phase I/II data concerning an oral HIF2α inhibitor, belzutifan, used for patients who experienced disease progression on IO/TKI therapy, has shown an ORR of 24% and a disease control rate of 80% across all risk groups [85]. A phase III trial of belzutifan monotherapy vs everolimus in previously treated patients is ongoing [86]. Synergistic effects have been observed in treatments with a combination of modalities; therefore, HIF2α inhibitors are being studied in combination to determine their efficacy and safety. A phase II study is investigating the combination of belzutifan with cabozantinib in patients who experienced disease progression after first- and second-line therapies. A separate phase III study is examining the efficacy of the combination of belzutifan with lenvatinib vs cabozantinib in patients with disease progression after first-line immunotherapy. The results from HIF2α inhibitor trials may add novel treatment options for patients with disease progression after immunotherapy and/or target therapy [87,88]. A three-arm phase III study aims to evaluate the efficacy and safety of pembrolizumab + belzutifan + lenvatinib or a coformulation of pembrolizumab and quavonlimab (CTLA4 inhibitor) + lenvatinib versus pembrolizumab + lenvatinib for patients with advanced ccRCC [89]. 

##### GAS6-AXL Pathway Inhibitor

AXL is a member of the TAM family together with the high-affinity ligand growth arrest-specific protein 6 (GAS6). The GAS6/AXL signaling pathway is associated with tumor cell growth, metastasis, invasion, angiogenesis, drug resistance, and immune regulation. In ccRCC, the constitutive expression of hypoxia-induced factor 1α leads to an increased expression of AXL. AXL overexpression has been associated with the development of resistance to VEGF inhibitors and the suppression of the innate immune response. Batiraxcept, a GAS6–AXL pathway inhibitor, had been tested in a phase Ib/II trial in combination with SOC drugs such as cabozantinib and nivolumab in patients with advanced ccRCC (NCT4300140) [84]. In this trial, early data have suggested that batiraxcept added to cabozantinib has no dose-limiting toxicities while showing some evidence of favorable clinical activity. The phase II portion of this study is currently open to recruitment [90].

### 3.2. nccRCC

#### 3.2.1. IO Doublet

CheckMate 920 is a multi-arm, phase IIIb/IV clinical trial of nivolumab plus ipilimumab treatment in patients with previously untreated advanced RCC and clinical features mostly excluded from the CheckMate214 study (i.e., non-clear cell RCC, brain metastases, and poor performance status). The primary endpoint was an incidence of grade ≥ 3 immune-mediated adverse events. The key secondary endpoints included PFS and the ORR, DOR, and TTF rates. The exploratory endpoints included OS rates [91].

#### 3.2.2. IO-TKI Combination (Triple Combination)

Several phase II trials are ongoing to elucidate the role of IO + TKIs in nccRCC. Enrollment has been completed for a highly anticipated phase II trial of triplet therapy with cabozantinib, nivolumab, and ipilimumab in nccRCC [92].

## 4. Discussion

Historically, RCC is known as an immuno-potent cancer with various clinical episodes (anecdotes) and has revealed a modest degree of susceptibility to cytokine therapy. The *VHL* gene, which is identified as the responsible gene in VHL disease, is a hallmark gene in sporadic ccRCC as well. Thus, the VEGF cascade has become a target for molecular therapies. The efficacy of some targeted drugs has been shown in key clinical trials. In admission, mTOR is also another target and mTORi, everolimus, and temsirolimus are introduced in the treatment of advanced and/or metastatic RCC. As it is true for various other cancers, based upon the molecular understanding of carcinogenesis and development, genetic alteration in cancer cells subcategorized as ‘driver mutation’ and ‘passenger mutation’ has led to the word ‘dirty cancer’, which is applied to cancers with a number of genetic changes. This type has the characteristic of being an inappropriate cancer type for targeted therapy. As for ccRCC, the new rationale using the blockade of immune checkpoint, including PD-1, PD-L1, and CTLA-4, was subjected to clinical trials. There have been drastic changes made to the treatment strategies for many types of cancer. Vigorous efforts towards exploring the specific key characteristics that determine favorable responses have not yet shown robust evidence besides PD-L1 expression. Nevertheless, the theoretical assumption, i.e., a higher mutation burden provides more neo antigens that might be responsible for tumor rejection, transforms ‘dirty cancer’ into a ‘favorable cancer’ for new immunotherapies. 

Discussing this issue from a pathological and molecular pathological point of view, although molecular profiles do not currently affect the care of patients with RCCs of any histological subtypes, emerging data from clinical trials of immuno-oncology agents combined with or compared to VEGF inhibitors suggest that distinct gene expression signatures reflecting a prominence of angiogenesis or immune infiltration correlate with the presence of sarcomatoid differentiation and response to therapy and could support personalized therapy choices in the future [17,30,93,94]. However, nccRCC is a very heterogeneous disease that can be further subdivided into more than 40 histological subtypes with clinical, histomorphological, immunohistochemical, and molecular features. These entities are emerging in the new 2022 World Health Organization (WHO) classification [64]. It should be noted that the tumor microenvironment-related gene expression signatures observed in nccRCCs may be due to different molecular pathways than those in ccRCC and may cause other hidden gene expression signatures, resulting in different therapeutic effects. The correlations between genomic alterations (such as the tumor mutation burden, neoantigen load, and chromosomal copy number alterations), the tumor microenvironment’s characteristics, and clinical response for target and IO therapies are under investigation [94,95].

Given the limited data on nccRCCs with a small number of cases in the previous studies, comprehensive clinicopathological and molecular genetic investigations with a large cohort for nccRCC patients at risk of metastasis are desirable for each specific nccRCC subtype in order to choose and develop more effective treatment strategies. Another major challenge in nccRCC is diagnosis. Due to the rarity of nccRCC, differential diagnosis remains difficult in some cases [96,97,98,99]. Notably, in a re-appraisal series of 33 cases diagnosed originally as so-called ‘unclassified’ RCC in patients aged 35 years or younger, 22 of 33 (66%) were reclassified as eosinophilic-solid and -cystic RCCs, FH-deficient RCCs, and succinate dehydrogenase-deficient RCCs [100]. Importantly, in clinical practice, the differential diagnosis between RCCs from UC, especially for FH-deficient RCCs, and high-grade distal nephron-related adenocarcinomas with overlapping morphology, including CDC and RMC with poorer prognoses, is sometimes extremely difficult but essential, nonetheless. In one large multi-institutional cohort study, 25% of cases initially diagnosed as potential CDCs were reclassified as FH-deficient RCC by immunostaining for FH and 2-succinocysteine [93]. A similar rate of reclassification from CDCs to FH-deficient RCCs or SMARCB1-deficient RMC occurred in a recent comprehensive genomic-profiling study detecting *FH* and *SMARCB1* mutations [101]. Combination therapy with conventional anticancer drugs remains the standard of care. Comprehensive pathological investigations are needed to choose appropriate treatments.

## 5. Conclusions

Regarding ccRCC, we have so many treatment options. One of the important questions is which of these options are appropriate: to combine these agents or to sequence them to maximize the outcome. We need good biomarkers for IO and/or target therapy to resolve certain issues. HIF inhibition is a novel, promising treatment target; several trials involving HIF2α inhibitors are ongoing. nccRCC are composed of various genetically and histologically different cancers. However, most of the active advancing prospective trials for patients with nccRCC emulate the developed regimens for ccRCC. Insights from molecular biology have helped elucidate oncogenic mechanisms, which fall into several subsets, based on biological characteristics. The treatments based on cancer histology and biology require further evidence regarding said characteristics.

## Figures and Tables

**Figure 1 biomedicines-10-02840-f001:**
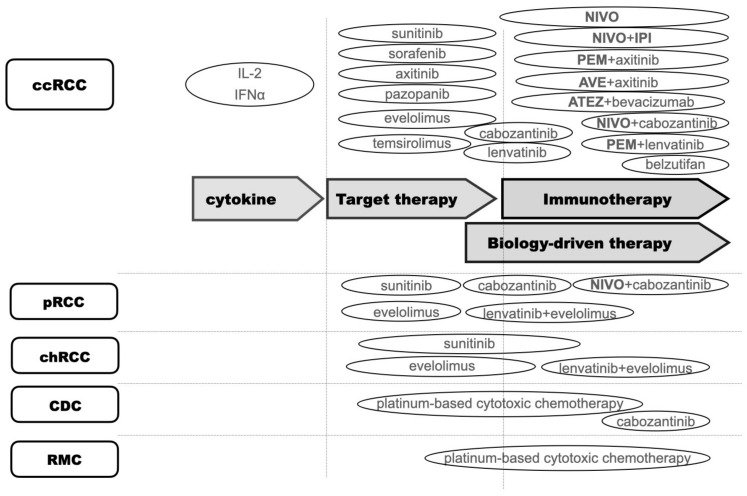
Timeline of systemic therapies for mRCC. Abbreviations: ccRCC, clear cell renal cell carcinoma; pRCC, papillary cell renal cell carcinoma; chRCC, chromophobe renal cell carcinoma; CDC, collecting duct carcinoma; RMC, SMARCB1-deficient renal medullary carcinoma; IFNα, interferon-alpha; IL-2, interleukin-2; NIVO, nivolumab; IPI, ipilimumab; PEM, pembrolizmab; AVE, avelumab; ATEZ, atezolizmab.

**Table 1 biomedicines-10-02840-t001:** Summary of efficacy in patients with ccRCC.

	ORR (%)	PFS (M)	OS (M)	RCT [Ref. Number]
Cytokine	14–23	3.3–5.5	17–18.8	EORTC GU 30012 [7]
Target therapy	28–40	8.3–12.3	34.3-N.R.	NCT00083889 [9]
				TARGET [10]
				AXIS [11]
				COMPARZ [12]
				CABOSUN [13]
				NCT01136733 [14]
				NCT00065468 [15]
				NCT00410124 [16]
IO doublet	39	12.2	55.7	Checkmate214 [17,18]
IO + VEGF inhibition	56–71	15.7–23.9	37.7-N.R.	Keynote426 [19]
				JAVELIN 101 [20]
				IMmotion151 [21]
				Checkmate9ER [22]
				CREAR [23]

Abbreviations; ccRCC, clear cell renal cell carcinoma; IO, immuno-oncology; VEGF, vascular endothelial growth factor; ORR, overall response rate; PFS, progression-free survival; M, month; OS, overall survival; RCT, randomized controlled trial; ref, reference; N.R., not reached.

**Table 2 biomedicines-10-02840-t002:** Summary of efficacy in patients with sarcomatoid ccRCC.

	ORR (%)	PFS (M)	OS (M)	RCT [Ref. Number]	Non-RCT [Ref. Number]
Cytokine	10	7.9	30.5	none	[29] (HD IL-2)
Target therapy	14–31.5	4.2–8.4	10.0-N.R.	Checkmate214 [30] Keynote426 [31] IMmotion151 [31] JAVELIN 101 [31] Checkmate9ER [32] CREAR [33]	[27,28]
IO doublet	61	26.5	N.R.	Checkmate214 [30,31]	
IO + VEGF inhibition	47–59	7.0-N.R.	N.R.	Keynote426 [31] IMmotion151 [31] JAVELIN 101 [31] Checkmate9ER [32] CREAR [33]	

Abbreviations; ccRCC, clear cell renal cell carcinoma; IO, immuno-oncology; VEGF, vascular endothelial growth factor; ORR, overall response rate; PFS, progression-free survival; M, month; OS, overall survival; RCT, randomized control trial; ref, reference; HD IL-2, High dose Interleukin-2; N.R., not reached.

**Table 3 biomedicines-10-02840-t003:** Summary of efficacy in patients with nccRCC.

	ORR(%)	PFS (M)	OS (M)	RCT [Ref. Number]	Non-RCT [Ref. number]
*pRCC*					
Cytokine	0–10.9	3.0–3.8	15.2–16.8	PERCY Quattro [34]	
Target therapy	5–15	5.7–9.2	16.6–21.4	SWOG 1107 [35] ARCC [36] ASPEN [37] ESPN [38] RAPTOR [39] SUPAP [40]	[41,42,43]
MET inhibitor	18–27	6.2–9.0	20-N.R.	CREATE [44] SAVOIR [45] PAPMET [46]	[47,48]
IO	8.3–28.8	4.9	12.3	Checkmate 374 [49]	Keynote427cohort B [50]
IO + MET inhibitor	29–47.5	4.9–12.5	12.3–28	CALYPSO [51]	NCT03635892 [52]
*FH-deficient RCC*					
Target therapy	64	21.1	not shown	none	AVATAR [53]
*chRCC*					
Target therapy	-44	8.9-N.R.	25.1–31.6	ASPEN [37] ESPN [38]	[43]
IO	9.5–28.5	not shown	not shown	Checkmate 374 [49]	Keynote427cohort B [50]
IO + VEGF inhibition	0–10	not shown	not shown	none	NCT03635892 [52]
*CDC*					
chemotherapy	26–39	7.1–8.8	10.5–12.5	none	[54,55,56]
Target therapy	0–35	3.1–4.0	7.0	none	[57] BONSAI [58]
*RMC*					
chemotherapy	18.8–29	2.8	8.1	none	[59,60]

Abbreviations; nccRCC, non-clear cell renal cell carcinoma; pRCC, papillary cell renal cell carcinoma; FH-deficient RCC, Fumarate hydratase-deficient renal cell carcinoma; chRCC, chromophobe renal cell carcinoma; IO, envat-oncology; VEGF, vascular endothelial growth factor; CDC, collecting duct carcinoma; RMC, SMARCB1-deficient renal medullary carcinoma; ORR, overall response rate; PFS, progression-free survival; M, month; OS, overall survival; RCT, randomized control trial; ref, reference; N.R., not reached.

## Data Availability

Data available in a publicly accessible repository.

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
