# Peer review of "Contemporary Drug Therapy for Renal Cell Carcinoma—Evidence Accumulation and Histological Implications in Treatment Strategy"

_biomedicines, 2022, doi:10.3390/biomedicines10112840_

Round 1

Reviewer 1 Report

I consider the manuscript to be sound. In my opinion, a further schematic figure, including the molecular pathways involved in renal cell carcinoma therapy, should be added to text, in order to improve the manuscript presentation.

Reviewer 2 Report

The authors sumarized the advances in drug therapy for renal cell carcinoma (RCC). I have two major concerns:

â‘ . Is it well-recognized to categorize RCC therapy into 3 (or 4?) eras (i.e. cytokine therapy, target therapy, and immuno-oncology (IO) therapy)? Especially concerning that it cannot be well distinguished among the so-called target therapy, IO therapy, and biology-driven therapy, as shown in Fig. 1.

â‘¡. Since there is not big difference among treatments on non-clear cell carcinoma (nccRCC) at the moment, it is redundant to review their therapies seperately. 

Besides,

â‘¢. Paragraphs on "ccRCC with sarcomatoid and/or rhabdoid differentiation" (starting on line 88, and line 359) are  quite abrupt, and logically it is not consistent with the parallel parts. 

â‘£. The "Conclusion" part is poorly sumarized. 

⑤. In line 15, "hat" is a misspelling.

Reviewer 3 Report

Manuscript entitled "Contemporary drug therapy for renal cell carcinoma; evidence accumulation and histological implications in treatment strategy" submitted for review in the journal Biomedicines provides an excellent review of clinical trials in the treatment strategy of renal cell carcinoma.

The authors have conducted a solid literature review and presented it in a clear manner for the audience. 

I think the article can be accepted for publication in its current form.

Author Response

We thank the reviewer for these excellent comments.

Round 2

Reviewer 2 Report

After re-evaluate the manuscript, I stick to my initial judgement that the overall quality is far from being accepted for publication. Generally, the whole logic, structure, and depth should be improved. In detail, the improvements should include, but not limit to: 

â‘  Abstract is poorly written. 

â‘¡ A numbering system for subheadings like (1.1, 1.1.1) can be used to make the whole structure much easier to follow.

â‘¢  Classification in Figure 1 is not consistent with the main text. For example, chemotherapy is discussed in the main text, while omitted in Fig. 1. Similarily, tRCC, FH-deficient RCC, etc. are not included in Fig. 1. 

â‘£ Some paragraphs are too long, and jumbled of several different aspects, which makes it diffcult to get the main information. 

⑤ Regarding to the studies mentioned in "Updated treatment strategy, Noteworthy clinical trials", outcomes (or conclusions) are not introduced.